# Risk Factors for Latent Tuberculosis Identified Using Epidemiological Investigation in Congregate Settings of Gyeongsan City, Republic of Korea (2014–2023)

**DOI:** 10.3390/pathogens14080740

**Published:** 2025-07-27

**Authors:** Seonyeong Park, Kwan Lee

**Affiliations:** 1Health and Public Administration Department, Republic Health Center, Gyeongsan 38616, Gyeongsangbuk-do, Republic of Korea; nurgong@korea.kr; 2Department of Preventive Medicine, Dongguk University College of Medicine, Gyeongju 38066, Gyeongsangbuk-do, Republic of Korea

**Keywords:** congregate settings, epidemiological investigation, latent tuberculosis, risk factor

## Abstract

Latent tuberculosis infection (LTBI) remains an important public health issue, as individuals can harbor *Mycobacterium tuberculosis* without symptoms and later develop active disease. This study aimed to assess the prevalence and risk factors associated with LTBI positivity among tuberculosis (TB) contacts in congregate settings in Gyeongsan City, the Republic of Korea (ROK), from 2014 to 2023. A total of 213 index cases and 3666 contacts were analyzed using data from the Korea Tuberculosis Infection Control System (KTB-NET). Overall, 20.7% of contacts tested positive for LTBI, with the highest rates observed among contacts aged ≥65 years (50.4%) and in healthcare facilities (34.8%). Binary logistic regression analyses revealed that age ≥65 years (OR: 2.93; 95% CI: 1.95–4.39; *p* < 0.001), social welfare facilities (OR: 2.75; 95% CI: 2.10–3.58; *p* < 0.001), workplaces (OR: 2.42; 95% CI: 1.88–3.10; *p* < 0.001), and healthcare facilities (OR: 3.42; 95% CI: 2.63–4.43; *p* < 0.001) were significantly associated with increased LTBI risk. These findings highlight the importance of targeted interventions and prevention strategies focused on older adults and high-risk groups to prevent future TB outbreaks by reducing the burden of LTBI.

## 1. Introduction

TB is an airborne infectious disease caused by *M. tuberculosis* [1]. LTBI is a state of persistent immune response to *M. tuberculosis* antigens without evidence of clinically manifested active TB [2], and on average, 5–10% of those who are infected will develop active TB disease over their lifetime [3]. In 2014, the global burden of LTBI was 23%, accounting for approximately 1.7 billion people [4].

According to the World Health Organization (WHO), approximately 8.6 million TB cases were reported globally in 2012, with Asia and Africa accounting for 58% and 27% of cases, respectively [5]; ROK had 39,545 TB cases (78.5 per 100,000 population) [6]. While TB incidence, prevalence, and mortality have declined over the past decade, elimination of TB remains unattainable without substantial resource investment. Despite government-led TB control policies in ROK, the country reported the second highest TB incidence and mortality rates among the Organization for Economic Co-operation and Development (OECD) member countries as of 2022, with 39 and 3.8 cases per 100,000 population, respectively [7]. This underscores the urgent need for a deeper understanding of TB dynamics within ROK, which could inform strategies applicable to both developed and developing nations. Epidemiological investigation of TB through KTB-NET is essential for understanding the distribution of associated risk factors [8]. In ROK, local public health centers play a crucial role in TB contact tracing, directly reaching out to individuals exposed to patients with TB [9]. These contacts, often belonging to congregate settings such as schools or workplaces, are classified as close or casual contacts based on infectivity, duration of exposure, and contact environment. This robust nationwide system enables effective TB testing and LTBI screening, which are key in identifying additional cases and curbing the spread of the disease.

This study aimed to identify risk factors for LTBI using data obtained through TB contact investigation conducted by the Gyeongsan City Public Health Center in ROK. We analyzed the demographic and clinical information of index cases and contact characteristics, including the types of congregate settings. Logistic regression analysis was used to identify risk factors for TB infection. Our findings can enhance the understanding of TB transmission dynamics and inform targeted interventions for better TB control.

## 2. Materials and Methods

### 2.1. Population Data

This study examined the prevalence of LTBI among 213 TB contact investigations that included the LTBI screening test, selected from a total of 308 TB epidemiological investigations conducted between January 2014 and December 2023. These investigations were included based on the availability of LTBI screening results, while 95 investigations were excluded owing to missing screening data (*n* = 50), duplicate records because of additional cases (*n* = 15), no contacts (*n* = 6), and epidemiological investigations conducted directly by medical hospitals (*n* = 24). Demographic and clinical data for the index cases were extracted from 213 investigations, including respiratory symptoms, cavitary lesions on chest radiography, and smear test results. Among the 8232 TB contacts, those who underwent only chest radiography without LTBI screening (*n* = 3319) and those who underwent a tuberculin skin test (TST) instead of IGRA (*n* = 880) were excluded. Additionally, individuals who had not undergone IGRA testing (*n* = 361) or had indeterminate IGRA results (*n* = 6) were also excluded. Consequently, 3666 contacts were included in the final analysis (Figure 1).

### 2.2. Measures and Definitions

Only contacts who underwent IGRA testing were included in the LTBI analysis. Contacts tested with TST were excluded due to the risk of false-positive results, as the BCG vaccination history was unclear. According to CDC guidelines, IGRA results (QuantiFERON-TB Gold In-Tube, QFT; SD BIOSENSOR, Suwon, Republic of Korea) with values ≥ 0.35 IU/mL were considered LTBI-positive, while these <0.35 IU/mL were classified as LTBI-negative. After excluding contacts with indeterminate or missing QFT results, those with valid results were classified as LTBI-positive or LTBI-negative. The proportion of positive cases was defined as the LTBI-positive rate.

Risk factors were analyzed using demographic variables (sex, age, and type of congregate settings) and clinical data (respiratory symptoms, cavitary on chest radiography, and sputum smear test results). Congregate settings were categorized as educational facilities, military camps, social welfare facilities, workplaces, and healthcare facilities. Educational facilities encompassed kindergartens, elementary schools, middle schools, high schools, and universities. Social welfare facilities included daycare centers for older adults or individuals with disabilities, nursing homes, and community welfare centers. Healthcare facilities comprised private clinics, long-term hospitals, psychiatric hospitals, and rehabilitation hospitals. Hospitals with ≥150 beds were excluded from the analysis, as they had dedicated infection control nurses and conducted their own TB epidemiological investigations.

In the context of TB epidemiological investigations, the extent of contact tracing is determined based on a comprehensive assessment of multiple factors, including the transmissibility and estimated duration of infectiousness of index cases, the duration and intensity of exposure, the nature of the environment in which the exposure occurred, and the vulnerability of the exposed individuals. Contact investigations prioritize individuals who are either (1) at high risk of TB infection or (2) at high risk of progression to active disease following infection. To estimate the infection period of index cases, clinical indicators such as the presence of TB-related symptoms, sputum smear results, and cavitary lesions on chest radiography were considered. TB-related symptoms included cough, sputum, hemoptysis, dyspnea, fever, and weight loss. These clinical features were incorporated as a composite variable to adjust for the infectious potential of index cases in the multivariable analysis.

Close contacts were defined as non-household individuals who had prolonged direct contact with an index case or had shared the same indoor environment. Transmission primarily occurs in confined indoor spaces, such as rooms and classrooms, whereas the likelihood of transmission in larger spaces, including large classrooms and corridors, is comparatively low. The temporal criterion for close contact is defined as either continuous (or daily) exposure exceeding ≥ 8 h per day in a confined space or a cumulative exposure ≥40 h during the estimated infection period. Nevertheless, based on the results of TB field investigations, individuals with an exposure duration below the standard time criteria may still be classified as close contacts. According to national TB control guidelines [10], examples of close contacts may also include individuals who share the same space for extended periods, engage in regular group activities, or have close interpersonal relationships.

### 2.3. Data Collection

This study analyzed data from 213 TB contact investigations conducted between January 2014 and December 2023. Demographic and clinical information for the index cases was retrieved from TB reports in KTB-NET (Available online: http://is.kdca.go.kr/ (accessed on 31 March 2025)). Corresponding data for the TB contacts were obtained to ensure a comprehensive dataset. All data were cleaned, indexed, and coded into structured Excel spreadsheets (Microsoft, Redmond, WA, USA) to facilitate subsequent analyses.

Although the KTB-NET database provided detailed information for index cases—including sex, age, diagnostic test results, clinical symptoms, prescribed medications, household composition and demographics, workplace information, height and weight, underlying diseases, smoking history, BCG vaccination status, and prior TB history—data for TB contacts were considerably limited. Available variables for contacts included only sex, age, and diagnostic method and results. While the role or status of contacts within congregate settings was recorded, the classification was often ambiguous. No information was available regarding symptoms, underlying conditions, BCG vaccination history, or previous TB infection among contacts.

### 2.4. Data Analysis

Data were exported from Excel to SPSS ver. 29.0 software (IBM Corp., Armonk, NY, USA) for analysis. The analysis comprised several steps. First, the general characteristics and risk factors of the index cases were summarized using means for continuous variables and frequencies (%) for categorical variables. Second, the LTBI positivity rate among contacts was calculated as frequencies and percentages. Third, binary logistic regression models were employed to assess associations between individual risk factors and LTBI positivity. Statistical significance was set at *p* < 0.05. Variables with *p* < 0.15 in the univariate analysis were initially selected, and variable selection was performed using stepwise selection. Multicollinearity among variables was assessed using the variance inflation factor (VIF), and all variables included had VIF values less than 2.0, indicating no significant multicollinearity. Model fit was evaluated using the Hosmer–Lemeshow goodness-of-fit test, and the explanatory power of the final model was reported using the Nagelkerke R^2^ statistic [11]. The regression model included variables such as respiratory symptoms, cavitary lesions on chest radiography, sputum smear results, sputum culture findings, Xpert results, sex, and age (categorized into three groups: <19, 19–64, and ≥65 years), along with congregate settings data.

### 2.5. Ethical Statement

This study protocol was exempted from ethical review by the Institutional Review Board of Dongguk University Gyeongju Hospital (IRB registration No. 110757-202505-HR-02-02) because the data analyzed did not contain information that could be used to identify individuals. De-identified data from TB reports in KTB-NET were used for the analysis. Consequently, the requirement for informed consent from human subjects was waived.

## 3. Results

### 3.1. Characteristics of Index Cases

The demographic and clinical characteristics of the index cases are summarized in Table 1. This study included 213 index cases, comprising 116 males (54.5%) and 97 females (45.5%). The age distribution was as follows: 17 cases (8.0%) aged <20 years, 56 (26.2%) aged 20–29 years, 21 (9.9%) aged 30–39 years, 17 (8.0%) aged 40–49 years, 21 (9.9%) aged 50–59 years, 13 (6.1%) aged 60–69 years, 17 (8.0%) aged 70–79 years, 45 (21.1%) aged 80–89 years, and 6 (2.8%) aged ≥90 years. Most cases (91.1%) were nationals of ROK, with only 19 (8.9%) foreign nationals. Among the congregate settings, educational facilities accounted for the highest number of cases (67; 31.4%), followed by workplaces (62; 29.1%) and social welfare facilities (44; 20.7%). Clinically, 150 index cases (70.4%) exhibited respiratory symptoms, and 54 (25.4%) had cavitary lesions on chest radiography. Furthermore, sputum smear tests were positive in 109 index cases (51.2%), indicating a substantial proportion of bacteriologically confirmed cases.

### 3.2. Characteristics of TB Contacts in Congregate Settings

We identified 3666 individuals as contacts of confirmed TB cases in congregate settings and designated them as candidates for LTBI evaluation (Table 2). The LTBI testing rate was 91.1%, and the positivity rate was 20.7%. When stratified by sex, 339 of 1750 male contacts (19.4%) tested positive, compared to 421 of 1916 female contacts (22.0%). The difference was not statistically significant (χ^2^ = 3.767; *p* = 0.052). Age-stratified analysis revealed a significant difference in LTBI positivity (χ^2^ = 183.27; *p* < 0.001). Among contacts aged <19 years, 70 (11.0%) tested positive, compared to 553 (20.1%) of those aged 19–64 years. Contacts aged ≥65 years showed a marked increase, with 137 (50.4%) testing positive and 135 (49.6%) testing negative, suggesting a strong age-dependent trend.

The LTBI positivity rate varied significantly across types of congregate settings (χ^2^ = 212.99; *p* < 0.001), with the highest rates observed in healthcare facilities (34.8%), followed by social welfare facilities (31.3%), workplaces (24.3%), and military camps (16.7%). Educational facilities had the lowest rate (11.0%).

Analysis of the positivity rate by the number of contacts revealed a statistically significant difference (χ^2^ = 52.3; *p* < 0.001). The positivity rate was lowest among groups with fewer than 10 contacts (18.6%) and highest among those with 30–49 contacts (25.9%). When stratified by the clinical characteristics of the index case, LTBI positivity rates were 19.6% for those with respiratory symptoms, 21.9% for those with cavitary lesions on chest radiography, and 21.4% for those with sputum smear positivity. However, none of these differences were statistically significant.

### 3.3. Risk Factors Associated with LTBI Positivity

Binary logistic regression analyses were conducted to identify significant risk factors for LTBI positivity based on demographic and clinical variables (Table 3). None of the clinical factors were significantly associated with LTBI positivity. However, age ≥ 65 years (odds ratio [OR]: 2.93; 95% confidence interval [CI]: 1.95–4.39; *p* < 0.001) and exposure in social welfare facilities (OR: 2.75; 95% CI: 2.10–3.58; *p* < 0.001), workplaces (OR: 2.42; 95% CI: 1.88–3.10; *p* < 0.001), and healthcare facilities (OR: 3.42; 95% CI: 2.63–4.43; *p* < 0.001) were identified as significant risk factors.

### 3.4. Temporal Trends in LTBI Positivity

To examine temporal trends in LTBI positivity among TB contacts, we reviewed annual positivity rates from 2014 to 2023 (Table 4). During this period, a total of 3666 contacts were tested, with the annual LTBI positivity rate ranging from 14.1% in 2020 to 32.3% in 2023. Notably, an increase was observed in 2023.

## 4. Discussion

This study presents the results of a population-based epidemiological analysis of TB in Gyeongsan City, a city in ROK with the second-highest incidence and mortality rates among OECD member nations. ROK manages TB patients in its jurisdiction at city and county public health centers and conducts a TB epidemiological investigation on congregate settings in the region. The Annual Report on the Notified TB in ROK—compiling data from KTB-NET and analyzing them by comparing them by metropolitan city and province, and by city and county and district—is published annually. The TB incidence rate in Gyeongsangbuk-do was 55.3 per 100,000 population, while that of Gyeongsan City was 42.8 per 100,000, both notably higher than the national incidence of 35.2 per 100,000 [12]. Gyeongsan City, located in Gyeongsangbuk-do, has the third-largest population in the province and a significant number of foreign workers, owing to the presence of both higher education institutions and industrial complexes. While some of these characteristics may be shared by other mid-sized cities in ROK, they do not fully represent the broader demographic or institutional landscape of the country; therefore, the generalizability of the findings to the entire ROK population may be limited. Although a nationwide survey was conducted on the prevalence of LTBI in the general population of ROK, studies based on city- or county-level TB contact epidemiological investigations have been lacking. This study addresses these gaps using data from mid-sized cities. Through comprehensive demographic and clinical data from 213 TB index cases and 3666 contacts, we aimed to identify major risk factors associated with LTBI positivity.

LTBI among contacts was diagnosed using IGRA, which demonstrates a sensitivity of 80–90% and a specificity greater than 95% in BCG-vaccinated populations. According to CDC guidelines, either TST or IGRA can be used to diagnose LTBI in immunocompetent adults [13,14]. In ROK, BCG vaccination was introduced in 1948 and was expanded into a nationwide program in 1952. Since 2008, BCG vaccination at four weeks of age has been recommended as part of the national childhood immunization schedule. Most major guidelines recommend the use of IGRA alone for LTBI screening in individuals with uncertain BCG vaccination status [15]. During data collection, no clinical information on contacts was available in the KTN-NET database. In addition, considering the potential for false-positive TST results due to prior BCG vaccination among Koreans, the limited recall of vaccination history among older adults, and the difficulty of verifying the BCG vaccination status among foreign nationals, this study excluded individuals who underwent LTBI screening using TST.

Various risk factors are known to influence the development of LTBI. Although household contacts were excluded from this study, it is well established that prolonged exposure to individuals with active TB in households or congregate settings significantly increases the risk of LTBI acquisition [16]. In our analysis, we observed a clear age-related trend observed in LTBI positivity, with individuals aged ≥65 years exhibiting significantly higher positivity rates than those aged <19 and 19–64 years. This finding aligns with previous studies reporting age-associated differences in TB prevalence both in ROK and internationally [17,18,19,20,21,22]. While a higher LTBI positivity rate was observed among male contacts in the present study, this difference was not statistically significant. Nonetheless, other investigations have also reported elevated LTBI positivity rates among males [15,23]. 

Substantial variation in LTBI positivity rates were observed across different types of congregate settings, with healthcare facilities showing the highest rates [13,20]. In our study, the healthcare facilities where public health centers conducted TB epidemiological investigations were long-term hospitals and psychiatric hospitals. Contacts within healthcare facilities were either patients or healthcare workers. Patients admitted to the same room as the index case were typically classified as close contacts. Healthcare workers included physicians, nurses, nursing assistants, nursing aides, caregivers, clinical laboratory technologists, radiologic technologists, and administrative staff. Among them, those with frequent ward-based contacts—such as nurses, nursing assistants, nursing aides, and caregivers—were usually classified as close contacts. In contrast, physicians in long-term hospitals were typically considered casual contacts due to limited exposure during brief ward rounds. However, physicians in psychiatric hospitals were often categorized as close contacts given their prolonged and repeated exposure to index cases in enclosed spaces such as consultation rooms. In ROK, LTBI screening is legally mandated for healthcare workers. Accordingly, individuals who had previously tested positive for LTBI through routine screening were excluded from contact investigations. As a result, LTBI status by occupational category could not be assessed. This underscores the urgent need for enhanced preventive measures and infection control strategies in healthcare facilities.

Binary logistic regression analysis revealed that none of the clinical characteristics of index cases, such as respiratory symptoms, cavitary lesions on chest radiography, and sputum smear positivity, were significantly associated with LTBI positivity among contacts. Although the infectiousness of index cases is a well-established factor in TB transmission [24,25], this null finding should be interpreted with particular caution. Several factors may account for this result. First, the unique epidemiological characteristics of South Korea, including the relatively high incidence of TB and prevalence of LTBI, may influence transmission dynamics. Second, our analysis could not distinguish between recent and past LTBI infections among contacts. Finally, environmental and structural conditions in congregate settings, such as ventilation and population density, may exert a stronger influence on LTBI risk than the clinical infectiousness of individual index cases. These considerations suggest that LTBI transmission in ROK is a multifactorial process requiring an integrated epidemiological and environmental approach.

In this study, among the demographic variables, age emerged as the most significant determinant of LTBI positivity, particularly among individuals aged ≥65 years. Additionally, the type of congregate setting was found to be an important factor; LTBI positivity was significantly higher in social welfare facilities, workplaces, and healthcare facilities compared to schools. These results may be attributed to the differences in age distribution across each setting. Schools and military camps (where military service is mandatory for males in ROK) are predominantly composed of younger individuals, whereas social welfare facilities, workplaces, and healthcare facilities tend to include a larger proportion of older adults. Given the increased incidence of active TB among older adults, particularly those aged ≥65 years, their presence in congregate settings may heighten the risk of both LTBI and subsequent progression to active TB [26]. This issue is particularly relevant in healthcare facilities, where the likelihood of contact with individuals with active TB is relatively high. These findings underscore the significance of demographic- and congregate setting-related factors in the transmission dynamics of TB and of its spread within communities. Moreover, the results provide a foundation for the development of targeted public health policies and interventions to reduce the risk of LTBI progression and transmission.

Although the primary objective of this study was to evaluate aggregate-level risk factors for LTBI, a descriptive review of annual IGRA positivity rates revealed considerable year-to-year variability, with rates ranging from 16% to 25% in most years. A notable increase was observed in 2023, when the positivity rate reached 32.3%. This marked rise may be attributed to a higher concentration of cases in high-risk congregate settings or an increased proportion of older adults contacts in that year. Nevertheless, the overall positivity rate aligns with previously reported findings. An IGRA-based cross-sectional study among healthcare workers in northern Peru reported a positivity rate of 17.9% (95% CI: 13.8–22.7%) [27], and a national survey in ROK reported a rate of 20.3% among adults [20]. These rates are consistent with the global LTBI prevalence of approximately 23%. Although formal time-series modeling could not be conducted due to the limited annual sample size, the observed temporal fluctuations underscore the need for future investigation incorporating time-series approaches to better elucidate changes in LTBI epidemiology over time.

This study had some limitations. First, chronic diseases that may influence the risk of LTBI were not classified, potentially affecting our results. In addition to age, diabetes, cancer, and immunosuppressive diseases are well-established risk factors for LTBI [28,29,30]. Although information on symptoms, smoking history, underlying diseases, height, and weight were available for TB index cases through the KTB-NET database, data for contacts were limited to sex, age, and the role or status within congregate settings during the epidemiological investigation. No data were available on symptoms or underlying diseases in contacts. With the 2025 update of KTB-NET, which includes additional fields such as symptoms, BCG vaccination history, and underlying diseases of contacts, future studies utilizing this expanded dataset will enable a more robust analysis of LTBI risk factors. We also found a significant association between certain types of congregate settings and LTBI positivity; however, these findings may be confounded by the age distribution within each setting. Due to limitations in sample size and available data, we were unable to perform stratified or interaction analyses to assess whether age modifies the relationship between types of congregate settings and LTBI positivity. Future research utilizing a larger, nationwide dataset is warranted to explore this relationship in greater depth. Although this study was conducted in a single city, our findings may still provide meaningful insights into TB transmission dynamics in comparable mid-sized urban settings in ROK.

Second, data collection was limited to a single public health center, which may have introduced sampling bias and limited the generalizability of our findings. Third, geographic factors, such as whether congregate settings were located in rural or urban areas, were not considered, despite their potential influence on TB transmission dynamics. As a result, the magnitude and impact of TB transmission may have been underestimated.

A particularly important limitation of this study is the inability to distinguish whether LTBI cases resulted from recent exposure to the index case identified during the contact investigation in determining whether LTBI-positive contacts acquired their infection from transmission pathways. This temporal ambiguity—namely, the uncertainty of whether LTBI-positive contacts acquired their infection from the index case or from prior exposure—substantially limits the ability to draw causal inferences about TB transmission pathways. Therefore, the observed association between the contact with the index case and LTBI positivity should be interpreted with caution, and this limitation should be emphasized more prominently when discussing the implications of the findings.

## 5. Conclusions

Our findings showed that, among TB contacts, age and facility type were the most important risk factors for LTBI positivity. Based on these findings, several public health recommendations can be proposed. First, targeted education and campaigns should be implemented to improve participation in LTBI screening and treatment, particularly among vulnerable populations. Second, mandatory pre-employment LTBI screening should be introduced for personnel working in nursing homes, considering the elevated risk identified in these facilities. Finally, the role of public health centers should be strengthened to facilitate active monitoring and follow-up of LTBI cases, thereby enhancing regional TB control efforts.

## Figures and Tables

**Figure 1 pathogens-14-00740-f001:**
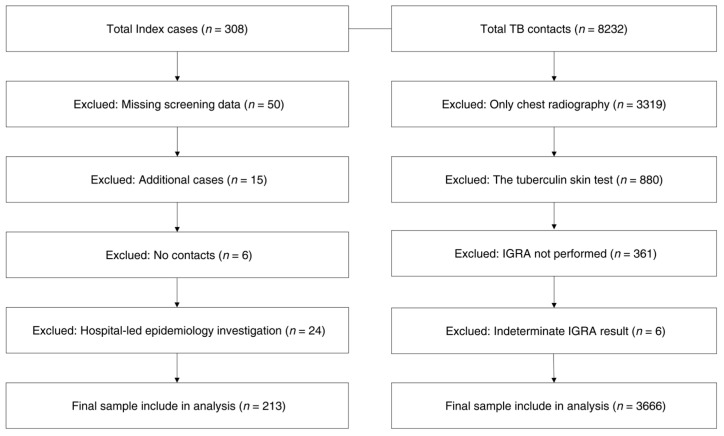
Algorithm for the selection of TB index cases who underwent TB epidemiological investigations and their contacts who were screened for LTBI using IGRA.

**Table 1 pathogens-14-00740-t001:** Demographic and clinical characteristics of 213 index cases.

Characteristics	Category	Number	Percentage (%)
Sex	Female	97	45.5
	Male	116	54.5
Age (years)	<20	17	8.0
	20–29	56	26.2
	30–39	21	9.9
	40–49	17	8.0
	50–59	21	9.9
	60–69	13	6.1
	70–79	17	8.0
	80–89	45	21.1
	≥90	6	2.8
Nationality	Republic of Korea national	194	91.1
	Foreign national	19	8.9
Types of congregate settings	Educational facilities	67	31.4
	Military camps	1	0.5
	Social welfare facilities	44	20.7
	Workplaces	62	29.1
	Healthcare facilities	39	18.3
Respiratory symptoms	Negative	63	29.6
	Positive	150	70.4
Cavity on chest radiography	Negative	159	74.6
	Positive	54	25.4
Sputum smear status	Negative	104	48.8
	Positive	109	51.2

**Table 2 pathogens-14-00740-t002:** Proportion of LTBI according to the characteristics of 3666 contacts.

Characteristics	Category	LTBI Positives (*n*, %)	LTBI Negatives (*n*, %)	Chi-Square (*p*-Value)
Sex	Female	339 (19.4)	1411 (80.6)	3.77 (0.052)
	Male	421 (22.0)	1495 (78.0)	
Age (years)	<19	70 (11.0)	569 (89.0)	183.27 (<0.001)
	19–64	553 (20.1)	2202 (79.9)	
	≥65	137 (50.4)	135 (49.6)	
Types of congregate settings	Educational facilities	118 (11.0)	1517 (89.0)	212.99 (<0.001)
	Military camps	13 (16.7)	65 (83.3)	
	Social welfare facilities	186 (31.3)	409 (68.7)	
	Workplaces	174 (24.3)	542 (75.7)	
	Healthcare facilities	199 (34.8)	373 (65.2)	
Number of contacts (cases)	<10	232 (18.6)	1018 (81.4)	52.3 (<0.001)
	10–29	104 (13.6)	662 (86.4)	
	30–49	320 (25.9)	916 (74.1)	
	≥50	104 (25.1)	310 (74.9)	
Respiratory symptoms	Negative	249 (22.6)	855 (77.4)	3.20 (0.074)
	Positive	511 (19.9)	2051 (80.1)	
Cavity on chest radiography	Negative	603 (20.4)	2346 (79.6)	0.74 (0.391)
	Positive	157 (21.9)	560 (78.1)	
Sputum smear status (PBS)	Negative	398 (20.2)	1573 (79.8)	0.74 (0.386)
	Positive	362 (21.4)	1333 (78.6)	
TB symptoms or signs in index *	Negative	139 (18.6)	608 (21.4)	2.57 (0.109)
	Positive	621 (21.3)	2298 (78.7)	
Total		760 (20.7)	2906 (79.3)	

* Positive if the TB index case has any of the following: respiratory symptoms, cavity on chest radiography, or positive sputum smear (PBS) status.

**Table 3 pathogens-14-00740-t003:** Binary logistic regression analysis of demographic and clinical variables for identifying risk factors for LTBI positivity.

Variable Type	Category	OR *	95% CI	*p*-Value
Age (years)	<19	1.0	Ref	
	19–64	1.09	0.81–1.48	0.574
	≥65	2.93	1.95–4.39	<0.001
Types of congregate settings	Educational facilities	1.0	Ref	
	Military camps	1.57	0.84–2.92	0.159
	Social welfare facilities	2.75	2.10–3.58	<0.001
	Workplaces	2.42	1.88–3.10	<0.001
	Healthcare facilities	3.42	2.63–4.43	<0.001

* Adjusted by sex, number of contacts, respiratory symptoms, and TB symptoms or signs in index.

**Table 4 pathogens-14-00740-t004:** Annual number of TB contacts and LTBI positive rate (2014–2023).

Year	No. of Contacts	LTBI-Positive	Positive Rate (%)
2014	370	78	21.1
2015	572	93	16.3
2016	293	66	22.5
2017	275	48	17.5
2018	210	48	19.2
2019	672	130	19.3
2020	269	38	14.1
2021	208	52	25.0
2022	376	71	18.9
2023	446	144	32.3
total	3666	760	20.7

## Data Availability

The data that support the findings of the study are available from the corresponding author upon reasonable request.

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
