# Peer review of "Risk Factors for Latent Tuberculosis Identified Using Epidemiological Investigation in Congregate Settings of Gyeongsan City, Republic of Korea (2014–2023)"

_pathogens, 2025, doi:10.3390/pathogens14080740_

Round 1
Reviewer 1 Report
Comments and Suggestions for Authors
This manuscript addresses an important public health issue — latent tuberculosis infection (LTBI) in congregate settings — using data from a regional TB control system in South Korea. The topic is timely and relevant, especially in the context of countries with intermediate to high TB burden. The manuscript is generally well-structured and contains clear tables and statistical analyses. However, there are several methodological and interpretative issues that require clarification or expansion.
Limited Generalizability of the Findings. The data were collected exclusively from one public health center in Gyeongsan, which limits the representativeness of the findings. The authors should explicitly discuss whether the demographic and institutional structure of this city is typical for South Korea or not.
Lack of Comorbidity Information. Chronic diseases such as diabetes, malignancies, or immunosuppressive conditions are important confounders in LTBI risk assessment, especially among older adults. The absence of such variables is acknowledged in the limitations, but the potential bias should be discussed in greater depth.
Temporal Ambiguity of LTBI Status. As the authors note, it is unclear whether LTBI-positive contacts acquired their infection from the index case or through previous exposure. This temporal ambiguity weakens causal inference and should be discussed more prominently in the limitations.
Unjustified Statistical Threshold (p < 0.15). The use of p < 0.15 as a threshold for variable selection in logistic regression requires justification, as it deviates from conventional norms. The authors should explain this decision or reconsider the approach to multivariable modeling.
Missing Flowchart of Study Inclusion and Exclusion. Although the exclusion of 95 epidemiological investigations is described, a flow diagram (e.g., based on STROBE recommendations) would greatly enhance transparency and reproducibility.
Under-discussed Non-significant Clinical Factors. Despite collecting rich clinical data (e.g., cavity presence, sputum smear results), none were associated with LTBI positivity in the multivariate model. This null finding deserves a more detailed discussion, as it contrasts with common assumptions regarding infectiousness indicators.
Vague Description of Contact Selection Criteria. The criteria for defining and selecting close contacts are briefly mentioned but not elaborated. The authors should clarify how exposure duration and environmental factors were operationalized in practice.
Reviewer 2 Report
Comments and Suggestions for Authors
I would like to thank you for the opportunity to review such an interesting and relevant paper. The article is very well structured and raises important data for public health assessment. Below, I presente small considerations and questions.
- I suggest updating reference 7 with data according to the latest WHO version of 2024.
- Positive LTBI was defined by individuals who used interferon-γ. Why didn't they use individuals who underwent TST? Even to carry out a comparative analysis.
- In the discussion, I suggest highlighting that the results found in the risk factors were in patients with LTBI who used interferon-γ and/or mentioning the percentage of reliability of the test for detecting LTBI.
Reviewer 3 Report
Comments and Suggestions for Authors
The positive results of IGRA test are mentioned as the basis of ITBL positive diagnosis
There were not mentioned the quantitative values of IGRA test
It was a calitative type of IGRA?
I think it could have been more relevant with a cut off value of IGRA positivity.
Reviewer 4 Report
Comments and Suggestions for Authors
The manuscript presents a relevant and timely study analyzing risk factors for latent tuberculosis infection (LTBI) in congregate settings in Korea. The study design is sound, using a substantial dataset from TB epidemiological investigations over a decade. However, while the paper has several strengths, it also exhibits important limitations in clarity, scientific rigor, and interpretation that should be addressed before publication.
Here are major comments with specific references to line numbers/sections in the manuscript:
Major Comments
- Study Representativeness & Generalizability (Section 2.1, Lines 63-68)
The study draws on data from a single public health center in Gyeongsan (Line 64), yet the conclusions are framed as generalizable to the national TB context. - Key Variables Missing in Risk Analysis (Table 3, Lines 174-176)
The risk factor analysis does not account for underlying comorbidities, BCG vaccination history, socioeconomic status, or smoking/alcohol exposure, all of which are relevant LTBI risk modifiers. - Ambiguity Around “Index Case Clinical Variables” (Lines 157-160; Table 2)
The analysis reports no significant association between the index case’s clinical features (e.g., respiratory symptoms, chest cavity, smear positivity) and LTBI rates in contacts. However, this interpretation may be misleading. Index case infectiousness is a well-established transmission factor. - Inconsistent and Ambiguous Terminology (Lines 71, 137, 138)
The term “LTBI candidates” is vague and non-standard. Similarly, “indicator case” (Line 18) is used instead of the standard term “index case.” - Regression Modeling Methodology Needs Clarification (Lines 103-108)
The use of p < 0.15 in univariate analysis to select variables for stepwise logistic regression is appropriate but not clearly explained. The model-building strategy lacks details (e.g., multicollinearity check, model fit statistics). - Lack of Temporal Analysis Despite 10-Year Dataset (Line 89; Study Period: 2014-2023)
The dataset spans a decade, yet no trends over time are discussed. Were LTBI positivity rates or risk factors consistent throughout this period? - Limited Discussion on Healthcare Worker Risk (Table 2, Line 150)
Healthcare facilities showed the highest LTBI rates (34.8%), yet this important finding is underexplored. Are these contacts patients or workers? - Age–Facility Confounding Not Explored (Lines 200-207)
The discussion notes that congregate setting effects may be confounded by age distribution (e.g., more older adults in welfare/healthcare settings), yet no stratified or interaction analysis is presented.
Minor Comments
- Define LTBI positivity clearly, including how indeterminate IGRA results were handled.
- Reformat tables for clarity—use consistent alignment, bold significant results, and clarify headers.
- Simplify the abstract language for better accessibility to a general audience.
- Remove redundant phrases and improve sentence flow, especially in the Discussion.
- Justify the use of p < 0.15 as the threshold for variable inclusion in regression models.
- Describe how missing or incomplete data were handled during analysis.
- Clearly define the composite “TB symptoms or signs” variable and explain its relevance.
- Strengthen the conclusion by adding concrete public health recommendations.
- Use consistent terminology throughout (e.g., "index case" vs. "indicator case").
- Perform a thorough grammar and language edit to correct minor errors and improve clarity.
The English in the manuscript is understandable, but it requires moderate to significant revision for grammar, clarity, and scientific tone. It’s not at the level expected for publication in a peer-reviewed international journal yet.
Round 2
Reviewer 1 Report
Comments and Suggestions for Authors
The manuscript has undergone significant improvement and is now suitable for publication. I recommend its acceptance in the current form.